# Analyzing Tree Architectures in Ensembles via Neural Tangent Kernel

**Ryuichi Kanoh[1,2], Mahito Sugiyama[1,2]**
[1]National Institute of Informatics
[2]The Graduate University for Advanced Studies, SOKENDAI
{kanoh, mahito}@nii.ac.jp

## Abstract

A *soft tree* is an actively studied variant of a decision tree that updates splitting rules using the gradient method. Although soft trees can take various architectures, their impact is not theoretically well known. In this paper, we formulate and analyze the *Neural Tangent Kernel* (NTK) induced by *soft tree ensembles* for arbitrary tree architectures. This kernel leads to the remarkable finding that only the number of leaves at each depth is relevant for the tree architecture in ensemble learning with an infinite number of trees. In other words, if the number of leaves at each depth is fixed, the training behavior in function space and the generalization performance are exactly the same across different tree architectures, even if they are not isomorphic. We also show that the NTK of asymmetric trees like decision lists does not degenerate when they get infinitely deep. This is in contrast to the perfect binary trees, whose NTK is known to degenerate and leads to worse generalization performance for deeper trees.

## 1 Introduction

*Ensemble learning* is one of the most important machine learning techniques used in real world applications. By combining the outputs of multiple predictors, it is possible to obtain robust results for complex prediction problems. *Decision trees* are often used as weak learners in ensemble learning (Breiman, 2001; Chen & Guestrin, 2016; Ke et al., 2017), and they can have a variety of structures such as various tree depths and whether or not the structure is symmetric. In the training process of tree ensembles, even a decision stump (Iba & Langley, 1992), a decision tree with the depth of 1, is known to be able to achieve zero training error as the number of trees increases (Freund & Schapire, 1996). However, generalization performance varies depending on weak learners (Liu et al., 2017), and the theoretical properties of their impact are not well known, which results in the requirement of empirical trial-and-error adjustments of the structure of weak learners.

In this paper, we focus on a *soft tree* (Kontschieder et al., 2015; Frosst & Hinton, 2017) as a weak learner. A soft tree is a variant of a decision tree that inherits characteristics of neural networks. Instead of using a greedy method (Quinlan, 1986; Breiman et al., 1984) to search splitting rules, soft trees make decision rules soft and simultaneously update the entire model parameters using the gradient method. Soft trees have been actively studied in recent years in terms of predictive performance (Kontschieder et al., 2015; Popov et al., 2020; Hazimeh et al., 2020), interpretability (Frosst & Hinton, 2017; Wan et al., 2021), and potential techniques in real world applications like pre-training and fine-tuning (Ke et al., 2019; Arik & Pfister, 2019). In addition, a soft tree can be interpreted as a Mixture-of-Experts (Jordan & Jacobs, 1993; Shazeer et al., 2017; Lepikhin et al., 2021), a practical technique for balancing computational cost and prediction performance.

To theoretically analyze soft tree ensembles, Kanoh & Sugiyama (2022) introduced the *Neural Tangent Kernel* (NTK) (Jacot et al., 2018) induced by them. The NTK framework analytically describes the behavior of ensemble learning with infinitely many soft trees, which leads to several non-trivial properties such as global convergence of training and the effect of parameter sharing in an oblivious tree (Popov et al., 2020; Prokhorenkova et al., 2018). However, their analysis is limited to a specific type of trees, perfect binary trees, and theoretical properties of other various types of tree architectures are still unrevealed.

Figure 1 illustrates representatives of tree architectures and their associated space partitioning in the case of a two-dimensional space. Note that each partition is not the axis parallel direction as we are considering soft trees. Not only symmetric trees, as shown in (a) and (b), but also asymmetric trees (Rivest, 1987), as shown in (c), are often used in practical applications (Tanno et al., 2019). Moreover, the structure in (d) corresponds to the *rule set ensembles* (Friedman & Popescu, 2008), a combination of rules to obtain predictions, which can be viewed as a variant of trees. Although each of these architectures has a different space partitioning and is practically used, it is not theoretically clear whether or not such architectures make any difference in the resulting predictive performance in ensemble learning.

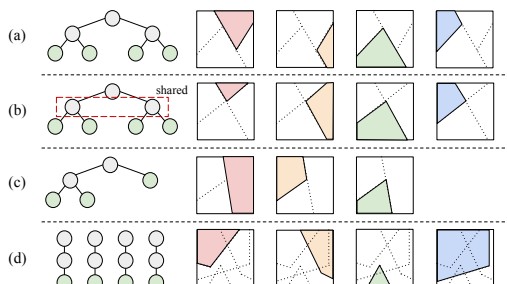

Figure 1: Schematic image of decision boundaries in an input space split by the (a) perfect binary tree, (b) oblivious tree, (c) decision list, and (d) rule set.

In this paper, we study the impact of *tree architectures* of soft tree ensembles from the NTK viewpoint. We analytically derive the NTK that characterizes the training behavior of soft tree ensemble with arbitrary tree architectures and theoretically analyze the generalization performance. Our contributions can be summarized as follows:

- **The NTK of soft tree ensembles is characterized by only the number of leaves per depth.**

  We derive the NTK induced by an infinite rule set ensemble (Theorem 2). Using this kernel, we obtain a formula for the NTK induced by an infinite ensemble of trees with arbitrary architectures (Theorem 3), which subsumes (Kanoh & Sugiyama, 2022, Theorem 1) as a special case (perfect binary trees). Interestingly, the kernel is determined by the number of leaves at each depth, which means that non-isomorphic trees can induce the same NTK (Corollary 1).

- **The decision boundary sharing does not affect to the generalization performance.**

  Since the kernel is determined by the number of leaves at each depth, infinite ensembles with trees and rule sets shown in Figure 1(a) and (d) induce exactly the same NTKs. This means that the way in which decision boundaries are shared does not change the model behavior within the limit of an infinite ensemble (Corollary 2).

- **The kernel degeneracy does not occur in deep asymmetric trees.**

  The NTK induced by perfect binary trees degenerates when the trees get deeper: the kernel values become almost identical for deep trees even if the inner products between input pairs are different, resulting in poor performance in numerical experiments. In contrast, we find that the NTK does not degenerate for trees that grow in only one direction (Proposition 1); hence generalization performance does not worsen even if trees become infinitely deep (Proposition 2, Figure 8).

## 2 PRELIMINARY

We formulate soft trees, which we use as weak learners in ensemble learning, and review the basic properties of the NTK and the existing result for the perfect binary trees.

### 2.1 SOFT TREES

Let us perform regression through an ensemble of $M$ soft trees. Given a data matrix $\boldsymbol{x} \in \mathbb{R}^{F \times N}$ composed of $N$ training samples $\boldsymbol{x}_1, \dots, \boldsymbol{x}_N$ with $F$ features, each weak learner, indexed by $m \in [M] = \{1, \dots, M\}$, has a parameter matrix $\boldsymbol{w}_m \in \mathbb{R}^{F \times \mathcal{N}}$ for internal nodes and $\boldsymbol{\pi}_m \in \mathbb{R}^{1 \times \mathcal{L}}$ for leaf nodes. They are defined in the following format:

$$\boldsymbol{x} = \left( \begin{array}{ccc} | & \cdots & | \\ \boldsymbol{x}_1 & \cdots & \boldsymbol{x}_N \\ | & \cdots & | \end{array} \right), \quad \boldsymbol{w}_m = \left( \begin{array}{ccc} | & \cdots & | \\ \boldsymbol{w}_{m,1} & \cdots & \boldsymbol{w}_{m,\mathcal{N}} \\ | & \cdots & | \end{array} \right), \quad \boldsymbol{\pi}_m = (\pi_{m,1}, \dots, \pi_{m,\mathcal{L}}),$$

where internal nodes and leaf nodes are indexed from 1 to $\mathcal{N}$ and 1 to $\mathcal{L}$, respectively. For simplicity, we assume that $\mathcal{N}$ and $\mathcal{L}$ are the same across different weak learners throughout the paper.

### 2.1.1 INTERNAL NODES

In a soft tree, the splitting operation at an intermediate node $n \in [\mathcal{N}] = \{1, \ldots, \mathcal{N}\}$ is not completely binary. To formulate the probabilistic splitting operation, we introduce the notation $\ell \swarrow n$ (resp. $n \searrow \ell$), which is a binary relation being true if a leaf $\ell \in [\mathcal{L}] = \{1, \ldots, \mathcal{L}\}$ belongs to the left (resp. right) subtree of a node $n$ and false otherwise. We also use an indicator function $\mathbb{1}_Q$ on the argument $Q$; that is, $\mathbb{1}_Q = 1$ if $Q$ is true and $\mathbb{1}_Q = 0$ otherwise. Every leaf node $\ell \in [\mathcal{L}]$ holds the probability that data reach to it, which is formulated as a function $\mu_{m,\ell} : \mathbb{R}^F \times \mathbb{R}^{F \times \mathcal{N}} \to [0,1]$ defined as

$$\mu_{m,\ell}(\boldsymbol{x}_i, \boldsymbol{w}_m) = \prod_{n=1}^{\mathcal{N}} \underbrace{\sigma(\boldsymbol{w}_{m,n}^\top \boldsymbol{x}_i)}_{\text{flow to the left}}^{\mathbb{1}_{\ell \swarrow n}} \underbrace{(1 - \sigma(\boldsymbol{w}_{m,n}^\top \boldsymbol{x}_i))}_{\text{flow to the right}}^{\mathbb{1}_{n \searrow \ell}}, \tag{1}$$

where $\sigma : \mathbb{R} \to [0,1]$ represents softened Boolean operation at internal nodes. The obtained value $\mu_{m,\ell}(\boldsymbol{x}_i, \boldsymbol{w}_m)$ is the probability of a sample $\boldsymbol{x}_i$ reaching a leaf $\ell$ in a soft tree $m$ with its parameter matrix $\boldsymbol{w}_m$. If the output of a decision function $\sigma$ takes only $0.0$ or $1.0$, this operation realizes the hard splitting used in typical decision trees. We do not explicitly use the bias term for simplicity as it can be technically treated as an additional feature.

Internal nodes perform as a sigmoid-like decision function such as the scaled error function $\sigma(p) = \frac{1}{2} \operatorname{erf}(\alpha p) + \frac{1}{2} = \frac{1}{2}(\frac{2}{\sqrt{\pi}} \int_0^{\alpha p} e^{-t^2} \, dt) + \frac{1}{2}$, the two-class sparsemax function $\sigma(p) = \operatorname{sparsemax}([\alpha p, 0])$ (Martins & Astudillo, 2016), or the two-class entmax function $\sigma(p) = \operatorname{entmax}([\alpha p, 0])$ (Peters et al., 2019). More precisely, any continuous function is possible if it is rotationally symmetric about the point $(0, 1/2)$ satisfying $\lim_{p \to \infty} \sigma(p) = 1$, $\lim_{p \to -\infty} \sigma(p) = 0$, and $\sigma(0) = 0.5$. Therefore, the theoretical results presented in this paper hold for a variety of sigmoid-like decision functions. When the scaling factor $\alpha \in \mathbb{R}^+$ (Frosst & Hinton, 2017) is infinitely large, sigmoid-like decision functions become step functions and represent the (hard) Boolean operation.

Equation 1 applies to arbitrary binary tree architectures. Moreover, if the flow to the right node $(1 - \sigma(\boldsymbol{w}_{m,n}^\top \boldsymbol{x}_i))$ is replaced with $0$, it is clear that the resulting model corresponds to a *rule set* (Friedman & Popescu, 2008), which can be represented as a linear graph. Note that the value $\sum_{\ell=1}^{\mathcal{L}} \mu_{m,\ell}(\boldsymbol{x}_i, \boldsymbol{w}_m)$ is always guaranteed to be $1$ for any soft trees, while it is not guaranteed for rule sets.

### 2.1.2 LEAF NODES

The prediction for each $\boldsymbol{x}_i$ from a weak learner $m$ parameterized by $\boldsymbol{w}_m$ and $\boldsymbol{\pi}_m$, represented as a function $f_m : \mathbb{R}^F \times \mathbb{R}^{F \times \mathcal{N}} \times \mathbb{R}^{1 \times \mathcal{L}} \to \mathbb{R}$, is given by

$$f_m(\boldsymbol{x}_i, \boldsymbol{w}_m, \boldsymbol{\pi}_m) = \sum_{\ell=1}^{\mathcal{L}} \pi_{m,\ell} \mu_{m,\ell}(\boldsymbol{x}_i, \boldsymbol{w}_m), \tag{2}$$

where $\pi_{m,\ell}$ denotes the response of a leaf $\ell$ of the weak learner $m$. This formulation means that the prediction output is the average of leaf values $\pi_{m,\ell}$ weighted by $\mu_{m,\ell}(\boldsymbol{x}_i, \boldsymbol{w}_m)$, the probability of assigning the sample $\boldsymbol{x}_i$ to the leaf $\ell$. In this model, $\boldsymbol{w}_m$ and $\boldsymbol{\pi}_m$ are updated during training with a gradient method. If $\mu_{m,\ell}(\boldsymbol{x}_i, \boldsymbol{w}_m)$ takes the value of only $1.0$ for one leaf and $0.0$ for the other leaves, the behavior of the soft tree is equivalent to a typical decision tree prediction.

### 2.1.3 AGGREGATION

When aggregating the output of multiple weak learners in ensemble learning, we divide the sum of the outputs by the square root of the number of weak learners, which results in

$$f(\boldsymbol{x}_i, \boldsymbol{w}, \boldsymbol{\pi}) = \frac{1}{\sqrt{M}} \sum_{m=1}^{M} f_m(\boldsymbol{x}_i, \boldsymbol{w}_m, \boldsymbol{\pi}_m). \tag{3}$$

This $1/\sqrt{M}$ scaling is known to be essential in the existing NTK literature to use the weak law of the large numbers (Jacot et al., 2018). Each of model parameters $\boldsymbol{w}_{m,n}$ and $\pi_{m,\ell}$ are initialized with zero-mean i.i.d. Gaussians with unit variances. We refer such an initialization as the *NTK initialization*.

## 2.2 NEURAL TANGENT KERNEL

For any learning model function $g$, the NTK induced by $g$ at a training time $\tau$ is formulated as a matrix $\widehat{\boldsymbol{H}}_\tau^* \in \mathbb{R}^{N \times N}$, in which each $(i, j) \in [N] \times [N]$ component is defined as

$$[\widehat{\boldsymbol{H}}_\tau^*]_{ij} := \widehat{\Theta}_\tau^*(\boldsymbol{x}_i, \boldsymbol{x}_j) := \left\langle \frac{\partial g(\boldsymbol{x}_i, \boldsymbol{\theta}_\tau)}{\partial \boldsymbol{\theta}_\tau}, \frac{\partial g(\boldsymbol{x}_j, \boldsymbol{\theta}_\tau)}{\partial \boldsymbol{\theta}_\tau} \right\rangle, \tag{4}$$

where $\widehat{\Theta}_\tau^* : \mathbb{R}^F \times \mathbb{R}^F \to \mathbb{R}$. The bracket $\langle \cdot, \cdot \rangle$ denotes the inner product and $\boldsymbol{\theta}_\tau \in \mathbb{R}^P$ is a concatenated vector of all trainable parameters at the training time $\tau$. An asterisk " $*$ " indicates that the model is arbitrary. The model function $g : \mathbb{R}^F \times \mathbb{R}^P \to \mathbb{R}$ used in Equation 4 is not limited to neural networks, and expected to be a variety of models. If we use soft trees introduced in Section 2.1 as weak learners, the NTK is formulated as $\sum_{m=1}^M \sum_{n=1}^{\mathcal{N}} \left\langle \frac{\partial f(\boldsymbol{x}_i, \boldsymbol{w}, \boldsymbol{\pi})}{\partial \boldsymbol{w}_{m,n}}, \frac{\partial f(\boldsymbol{x}_j, \boldsymbol{w}, \boldsymbol{\pi})}{\partial \boldsymbol{w}_{m,n}} \right\rangle +$ $\sum_{m=1}^M \sum_{\ell=1}^{\mathcal{L}} \left\langle \frac{\partial f(\boldsymbol{x}_i, \boldsymbol{w}, \boldsymbol{\pi})}{\partial \pi_{m,\ell}}, \frac{\partial f(\boldsymbol{x}_j, \boldsymbol{w}, \boldsymbol{\pi})}{\partial \pi_{m,\ell}} \right\rangle.$

If the NTK does not change from its initial value during training, one could describe the behavior of functional gradient descent with an infinitesimal step size under the squared loss using kernel ridgeless regression with the NTK (Jacot et al., 2018; Lee et al., 2019), which leads to the theoretical understanding of the training behavior. Such a property gives us a data-dependent generalization bound (Bartlett & Mendelson, 2003), which is important in the context of over-parameterization. The kernel does not change from its initial value during the gradient descent with an infinitesimal step size when considering an infinite width neural network (Jacot et al., 2018) or an infinite number of soft perfect binary trees (Kanoh & Sugiyama, 2022) under the NTK initialization. Models with the same *limiting NTK*, which is the NTK induced by a model with infinite width or infinitely many weak learners, have exactly equivalent training behavior in function space.

The NTK induced by a soft tree ensemble with infinitely many perfect binary trees, that is, the NTK when $M \to \infty$, is known to be obtained in closed-form at initialization:

**Theorem 1** (Kanoh & Sugiyama (2022)). *Let $\boldsymbol{u} \in \mathbb{R}^F$ be any column vector sampled from zero-mean i.i.d. Gaussians with unit variance. The NTK for an ensemble of soft perfect binary trees with tree depth $D$ converges in probability to the following deterministic kernel as $M \to \infty$,*

$$\Theta^{(D,\mathrm{PB})}(\boldsymbol{x}_i, \boldsymbol{x}_j) := \lim_{M \to \infty} \widehat{\Theta}_0^{(D,\mathrm{PB})}(\boldsymbol{x}_i, \boldsymbol{x}_j)$$
$$= \underbrace{2^D D \, \Sigma(\boldsymbol{x}_i, \boldsymbol{x}_j)(\mathcal{T}(\boldsymbol{x}_i, \boldsymbol{x}_j))^{D-1} \dot{\mathcal{T}}(\boldsymbol{x}_i, \boldsymbol{x}_j)}_{\text{contribution from internal nodes}} + \underbrace{(2\mathcal{T}(\boldsymbol{x}_i, \boldsymbol{x}_j))^D}_{\text{contribution from leaves}}, \tag{5}$$

*where $\Sigma(\boldsymbol{x}_i, \boldsymbol{x}_j) := \boldsymbol{x}_i^\top \boldsymbol{x}_j$, $\mathcal{T}(\boldsymbol{x}_i, \boldsymbol{x}_j) := \mathbb{E}[\sigma(\boldsymbol{u}^\top \boldsymbol{x}_i)\sigma(\boldsymbol{u}^\top \boldsymbol{x}_j)]$, and $\dot{\mathcal{T}}(\boldsymbol{x}_i, \boldsymbol{x}_j) := \mathbb{E}[\dot{\sigma}(\boldsymbol{u}^\top \boldsymbol{x}_i)\dot{\sigma}(\boldsymbol{u}^\top \boldsymbol{x}_j)]$. Moreover, when the decision function is the scaled error function, $\mathcal{T}(\boldsymbol{x}_i, \boldsymbol{x}_j)$ and $\dot{\mathcal{T}}(\boldsymbol{x}_i, \boldsymbol{x}_j)$ are analytically obtained in the closed-form as*

$$\mathcal{T}(\boldsymbol{x}_i, \boldsymbol{x}_j) = \frac{1}{2\pi} \arcsin\left( \frac{\alpha^2 \Sigma(\boldsymbol{x}_i, \boldsymbol{x}_j)}{\sqrt{(\alpha^2 \Sigma(\boldsymbol{x}_i, \boldsymbol{x}_i) + 0.5)(\alpha^2 \Sigma(\boldsymbol{x}_j, \boldsymbol{x}_j) + 0.5)}} \right) + \frac{1}{4}, \tag{6}$$

$$\dot{\mathcal{T}}(\boldsymbol{x}_i, \boldsymbol{x}_j) = \frac{\alpha^2}{\pi} \frac{1}{\sqrt{(1 + 2\alpha^2 \Sigma(\boldsymbol{x}_i, \boldsymbol{x}_i))(1 + 2\alpha^2 \Sigma(\boldsymbol{x}_j, \boldsymbol{x}_j)) - 4\alpha^4 \Sigma(\boldsymbol{x}_i, \boldsymbol{x}_j)^2}}. \tag{7}$$

Here, "PB" stands for a "P"erfect "B"inary tree. The dot used in $\dot{\sigma}(\boldsymbol{u}^\top \boldsymbol{x}_i)$ means the first derivative, and $\mathbb{E}[\cdot]$ means the expectation. The scalar $\pi$ in Equation 6 and Equation 7 is the circular constant, and $\boldsymbol{u}$ corresponds to $\boldsymbol{w}_{m,n}$ at any internal nodes. We can derive the formula of the limiting kernel by treating the number of trees in a tree ensemble like the width of the neural network, although the neural network and the soft tree ensemble appear to be different models.

## 3 THEORETICAL RESULTS

We first consider rule set ensembles shown in Figure 1(d) and provide its NTK in Section 3.1. This becomes the key component to introduce the NTKs for trees with arbitrary architectures in Section 3.2. Due to space limitations, detailed proofs are given in the Appendix.

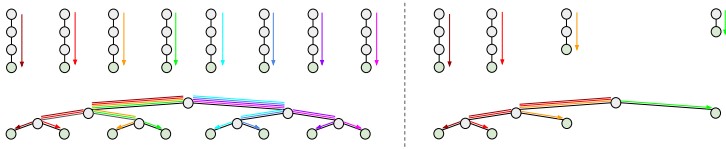

Figure 2: Correspondence between rule sets and binary trees. The top shows the corresponding rule sets for the bottom tree architectures.

## 3.1 NTK for Rule Sets

We prove that the NTK induced by a rule set ensemble is obtained in the closed-form as $M \to \infty$ at initialization:

**Theorem 2.** *The NTK for an ensemble of $M$ soft rule sets with the depth $D$ converges in probability to the following deterministic kernel as $M \to \infty$,*

$$
\begin{aligned}
\Theta^{(D,\text{Rule})}(\boldsymbol{x}_i, \boldsymbol{x}_j) &:= \lim_{M \to \infty} \widehat{\Theta}_0^{(D,\text{Rule})}(\boldsymbol{x}_i, \boldsymbol{x}_j) \\
&= \underbrace{D\, \Sigma(\boldsymbol{x}_i, \boldsymbol{x}_j)(\mathcal{T}(\boldsymbol{x}_i, \boldsymbol{x}_j))^{D-1}\dot{\mathcal{T}}(\boldsymbol{x}_i, \boldsymbol{x}_j)}_{\text{contribution from internal nodes}} + \underbrace{(\mathcal{T}(\boldsymbol{x}_i, \boldsymbol{x}_j))^D}_{\text{contribution from leaves}}.
\end{aligned}
\tag{8}
$$

We can see that the limiting NTK induced by an infinite ensemble of $2^D$ rules coincides with the limiting NTK of the perfect binary tree in Theorem 1: $2^D \Theta^{(D,\text{Rule})}(\boldsymbol{x}_i, \boldsymbol{x}_j) = \Theta^{(D,\text{PB})}(\boldsymbol{x}_i, \boldsymbol{x}_j)$. Here, $2^D$ corresponds to the number of leaves in a perfect binary tree. Figure 2 gives us an intuition: by duplicating internal nodes, we can always construct rule sets that correspond to a given tree by decomposing paths from the root to leaves, where the number of rules in the rule set corresponds to the number of leaves in the tree.

## 3.2 NTK for Trees with Arbitrary Architectures

Using our interpretation that a tree is a combination of multiple rule sets, we generalize Theorem 1 to include arbitrary architectures such as an asymmetric tree shown in the right panel of Figure 2.

**Theorem 3.** *Let $\mathcal{Q} : \mathbb{N} \to \mathbb{N} \cup \{0\}$ be a function that receives any depth and returns the number of leaves connected to internal nodes at the input depth. For any tree architecture, the NTK for an ensemble of soft trees converges in probability to the following deterministic kernel as $M \to \infty$,*

$$
\Theta^{(\text{ArbitraryTree})}(\boldsymbol{x}_i, \boldsymbol{x}_j) := \lim_{M \to \infty} \widehat{\Theta}_0^{(\text{ArbitraryTree})}(\boldsymbol{x}_i, \boldsymbol{x}_j) = \sum_{d=1}^{D} \mathcal{Q}(d)\, \Theta^{(d,\text{Rule})}(\boldsymbol{x}_i, \boldsymbol{x}_j).
\tag{9}
$$

We can see that this formula covers the limiting NTK for perfect binary trees $2^D \Theta^{(D,\text{Rule})}(\boldsymbol{x}_i, \boldsymbol{x}_j)$, as a special case by letting $\mathcal{Q}(D) = 2^D$ and 0 otherwise.

Kanoh & Sugiyama (2022) used mathematical induction to prove Theorem 1. However, this technique is limited to perfect binary trees. Consequently, we have now invented an alternative way of deriving the limiting NTK: treating a tree as a combination of independent rule sets using the symmetric properties of the decision function and the statistical independence of the leaf parameters.

It is also possible to show that the limiting kernel does not change during training:

**Theorem 4.** *Let $\lambda_{\min}$ and $\lambda_{\max}$ be the minimum and maximum eigenvalues of the limiting NTK. Assume $\|\boldsymbol{x}_i\|_2 = 1$ for all $i \in [N]$ and $\boldsymbol{x}_i \neq \boldsymbol{x}_j$ $(i \neq j)$. For ensembles of arbitrary soft trees with the NTK initialization trained under gradient flow with a learning rate $\eta < 2/(\lambda_{\min} + \lambda_{\max})$ and a positive finite scaling factor $\alpha$, we have, with high probability,*

$$
\sup \left| \widehat{\Theta}_\tau^{(\text{ArbitraryTree})}(\boldsymbol{x}_i, \boldsymbol{x}_j) - \widehat{\Theta}_0^{(\text{ArbitraryTree})}(\boldsymbol{x}_i, \boldsymbol{x}_j) \right| = \mathcal{O}\left(\frac{1}{\sqrt{M}}\right).
\tag{10}
$$

Therefore, we can analyze the training behavior based on kernel regression.

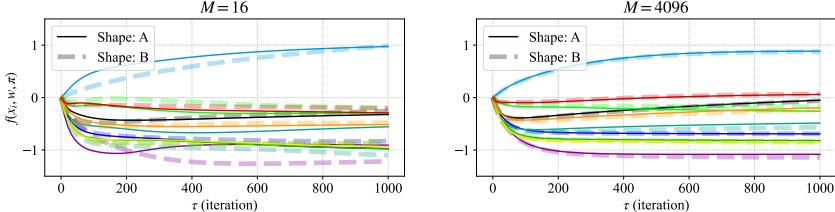

Figure 3: Non-isomorphic tree architectures used in ensembles that induce the same limiting NTK.

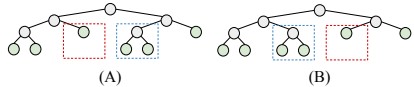

Figure 4: Output dynamics for test data points. Each line color corresponds to each data point.

Each rule set corresponds to a path to a leaf in a tree, as shown in Figure 2. Therefore, the depth of a rule set corresponds to the depth at which a leaf is present. Since Theorem 3 tells us that the limiting NTK depends on only the number of leaves at each depth with respect to tree architecture, the following holds:

**Corollary 1.** *The same limiting NTK can be induced from trees that are not isomorphic.*

For example, for two trees illustrated in Figure 3, $\mathcal{Q}(1) = 0$, $\mathcal{Q}(2) = 2$, and $\mathcal{Q}(3) = 4$. Therefore, the limiting NTKs are identical for ensembles of these trees and become $2\Theta^{(2,\text{Rule})}(\boldsymbol{x}_i, \boldsymbol{x}_j) + 4\Theta^{(3,\text{Rule})}(\boldsymbol{x}_i, \boldsymbol{x}_j)$. Since they have the same limiting NTKs, their training behaviors in function space and generalization performances are exactly equivalent when we consider infinite ensembles, although they are not isomorphic and were expected to have different properties.

To see this phenomenon empirically, we trained two types of ensembles; one is composed of soft trees in the left architecture in Figure 3 and the other is in the right-hand-side architecture in Figure 3. We tried two settings, $M = 16$ and $M = 4096$, to see the effect of the number of trees (weak learners). The decision function is a scaled error function with $\alpha = 2.0$. Figure 4 shows trajectories during full-batch gradient descent with a learning rate of $0.1$. Outputs at initialization are shifted to zero (Chizat et al., 2019). There are 10 randomly generated training points and 10 randomly generated test data points with dimension $F = 5$. Each line corresponds to each data point, and solid and dotted lines denote ensembles of left and right architecture, respectively. This result shows that two trajectories (solid and dotted lines for each color) become similar if $M$ is large, meaning that the property shown in Corollary 1 is empirically effective.

When we compare a rule set and a tree under the same number of leaves as shown in Figure 1(a) and (d), it is clear that the rule set has a larger representation power as it has more internal nodes and no decision boundaries are shared. However, when the collection of paths from the root to leaves in a tree is the same as the corresponding rule set as shown in Figure 2, their limiting NTKs are equivalent. Therefore, the following corollary holds:

**Corollary 2.** *Sharing of decision boundaries through parameter sharing does not affect the limiting NTKs.*

This result generalizes the result in (Kanoh & Sugiyama, 2022), which shows that the kernel induced by an oblivious tree, as shown in Figure 1(b), converges to the same kernel induced by a non-oblivious one, as shown in Figure 1(a), in the limit of infinite trees.

## 4 CASE STUDY: DECISION LIST

As a typical example of asymmetric trees, we consider a tree that grows in only one direction, as shown in Figure 5, often called a *decision list* (Rivest, 1987) and commonly used in practical applications (Letham et al., 2015). In this architecture, one leaf exists at each depth, except for leaves at the final depth, where there are two leaves.

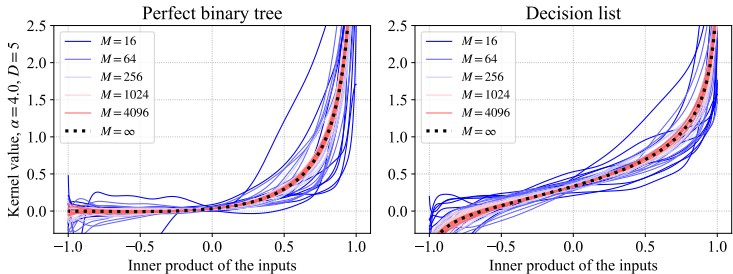

Figure 5: Decision list: a binary tree that grows in only one direction.

Figure 6: An empirical demonstration for (Left) perfect binary tree and (Right) decision list ensembles on the convergence of $\widehat{\Theta}_0^{(5,\mathrm{PB})}(\boldsymbol{x}_i, \boldsymbol{x}_j)$ and $\widehat{\Theta}_0^{(5,\mathrm{DL})}(\boldsymbol{x}_i, \boldsymbol{x}_j)$ to the fixed limit $\Theta^{(5,\mathrm{PB})}(\boldsymbol{x}_i, \boldsymbol{x}_j)$ and $\Theta^{(5,\mathrm{DL})}(\boldsymbol{x}_i, \boldsymbol{x}_j)$ as $M$ increases. The kernel induced by finite trees is numerically calculated and plotted 10 times with parameter re-initialization.

## 4.1 NTK FOR DECISION LISTS

We show that the NTK induced by decision lists is formulated in closed-form as $M \to \infty$ at initialization:

**Proposition 1.** *The NTK for an ensemble of soft decision lists with the depth $D$ converges in probability to the following deterministic kernel as $M \to \infty$,*

$$
\begin{aligned}
\Theta^{(D,\mathrm{DL})}(\boldsymbol{x}_i, \boldsymbol{x}_j) &:= \lim_{M \to \infty} \widehat{\Theta}_0^{(D,\mathrm{DL})}(\boldsymbol{x}_i, \boldsymbol{x}_j) \\
&= \Theta^{(1,\mathrm{Rule})}(\boldsymbol{x}_i, \boldsymbol{x}_j) + \Theta^{(2,\mathrm{Rule})}(\boldsymbol{x}_i, \boldsymbol{x}_j) + \cdots + 2\Theta^{(D,\mathrm{Rule})}(\boldsymbol{x}_i, \boldsymbol{x}_j) \\
&= \underbrace{\Sigma\left(\boldsymbol{x}_i, \boldsymbol{x}_j\right) \dot{\mathcal{T}}\left(\boldsymbol{x}_i, \boldsymbol{x}_j\right) \left( \sum_{d=1}^{D} \left( d\left(\mathcal{T}\left(\boldsymbol{x}_i, \boldsymbol{x}_j\right)\right)^{d-1} \right) + D\left(\mathcal{T}\left(\boldsymbol{x}_i, \boldsymbol{x}_j\right)\right)^{D-1} \right)}_{\text{contribution from internal nodes}} \\
&\quad + \underbrace{\sum_{d=1}^{D} \left( \left(\mathcal{T}\left(\boldsymbol{x}_i, \boldsymbol{x}_j\right)\right)^d \right) + \left(\mathcal{T}\left(\boldsymbol{x}_i, \boldsymbol{x}_j\right)\right)^D}_{\text{contribution from leaves}}.
\end{aligned}
\tag{11}
$$

In Proposition 1, "DL" stands for a "D"ecision "L"ist. The first equation comes from Theorem 3.

We numerically demonstrate the convergence of the kernels for perfect binary trees and decision lists in Figure 6 when the number $M$ of trees gets larger. We use two simple inputs: $\boldsymbol{x}_i = \{1, 0\}$ and $\boldsymbol{x}_j = \{\cos(\beta), \sin(\beta)\}$ with $\beta = [0, \pi]$. The scaled error function is used as a decision function. The kernel induced by finite trees is numerically calculated 10 times with parameter re-initialization for each of $M = 16, 64, 256, 1024,$ and 4096. We empirically observe that the kernels induced by sufficiently many soft trees converge to the limiting kernel given in Equation 5 and Equation 11 shown by the dotted lines in Figure 6. The kernel values induced by a finite ensemble are already close to the limiting NTK if the number of trees is larger than several hundred, which is a typical order of the number of trees in practical applications (Popov et al., 2020). This indicates that our NTK analysis is also effective in practical applications with finite ensembles.

## 4.2 DEGENERACY

Next, we analyze the effect of the tree depth to the kernel values. It is known that overly deep soft perfect binary trees induce the *degeneracy* phenomenon (Kanoh & Sugiyama, 2022), and we

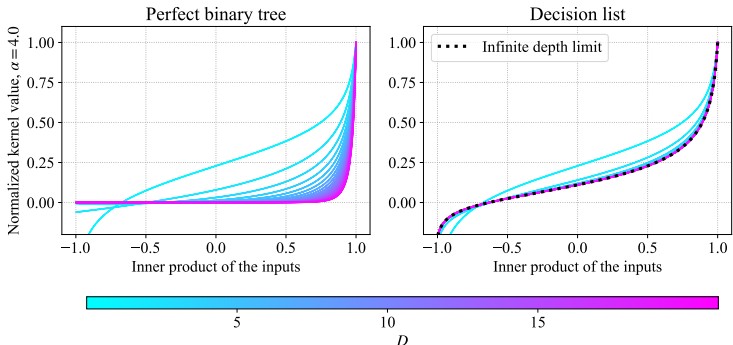

Figure 7: Depth dependency of (Left) $\Theta^{(D,\mathrm{PB})}(\boldsymbol{x}_i, \boldsymbol{x}_j)$ and (Right) $\Theta^{(D,\mathrm{DL})}(\boldsymbol{x}_i, \boldsymbol{x}_j)$. For decision lists, the limit of infinite depth is indicated by the dotted line.

analyzed whether or not this phenomenon also occurs in asymmetric trees like decision lists. Since $0 < \mathcal{T}(\boldsymbol{x}_i, \boldsymbol{x}_j) < 0.5$, replacing the summation in Equation 11 with an infinite series, we can obtain the closed-form formula when the depth $D \to \infty$ in the case of decision lists:

**Proposition 2.** *The NTK for an ensemble of soft decision lists with an infinite depth converges in probability to the following deterministic kernel as $M \to \infty$,*

$$\lim_{D \to \infty} \Theta^{(D,\mathrm{DL})}(\boldsymbol{x}_i, \boldsymbol{x}_j) = \underbrace{\frac{\Sigma(\boldsymbol{x}_i, \boldsymbol{x}_j)\,\dot{\mathcal{T}}(\boldsymbol{x}_i, \boldsymbol{x}_j)}{(1 - \mathcal{T}(\boldsymbol{x}_i, \boldsymbol{x}_j))^2}}_{\text{contribution from internal nodes}} + \underbrace{\frac{\mathcal{T}(\boldsymbol{x}_i, \boldsymbol{x}_j)}{1 - \mathcal{T}(\boldsymbol{x}_i, \boldsymbol{x}_j)}}_{\text{contribution from leaves}} . \tag{12}$$

Thus the limiting NTK $\Theta^{(D,\mathrm{DL})}$ of decision lists neither degenerates nor diverges as $D \to \infty$.

Figure 7 shows how the kernel changes as depth changes. In the case of the perfect binary tree, the kernel value sticks to zero as the inner product of the input gets farther from $1.0$ (Kanoh & Sugiyama, 2022), whereas in the decision list case, the kernel value does not stay at zero. In other words, deep perfect binary trees cannot distinguish between vectors with a 90-degree difference in angle and vectors with a 180-degree difference in angle. Meanwhile, even if the decision list becomes infinitely deep, the kernel does not degenerate as shown by the dotted line in the right panel of Figure 7. This implies that a deterioration in generalization performance is not likely to occur even if the model gets infinitely deep. We can understand such behavior intuitively from the following reasoning. When the depth of the perfect binary tree is infinite, all splitting regions become infinitely small, meaning that every data point falls into a unique leaf. In contrast, when a decision list is used, large splitting regions remain, so not all data are separated. This can avoid the phenomenon of separating data being equally distant.

### 4.3 NUMERICAL EXPERIMENTS

We experimentally examined the effects of the degeneracy phenomenon discussed in Section 4.2.

**Setup.** We used 90 classification tasks in the UCI database (Dua & Graff, 2017), each of which has fewer than 5000 data points as in (Arora et al., 2020). We performed kernel regression using the limiting NTK defined in Equation 5 and Equation 11, equivalent to the infinite ensemble of the perfect binary trees and decision lists. We used $D$ in $\{2, 4, 8, 16, 32, 64, 128\}$ and $\alpha$ in $\{1.0, 2.0, 4.0, 8.0, 16.0, 32.0\}$. The scaled error function is used as a decision function. To consider the ridge-less situation, regularization strength is fixed to $1.0 \times 10^{-8}$. We report four-fold cross-validation performance with random data splitting as in Arora et al. (2020) and Fernández-Delgado et al. (2014). Other details are provided in the Appendix.

**Performance.** Figure 8 shows the averaged performance in classification accuracy on 90 datasets. The generalization performance decreases as the tree depth increases when perfect binary trees are used as weak learners. However, no significant deterioration occurs when decision lists are used as weak learners. This result is consistent with the degeneracy properties as discussed in Section 4.2.

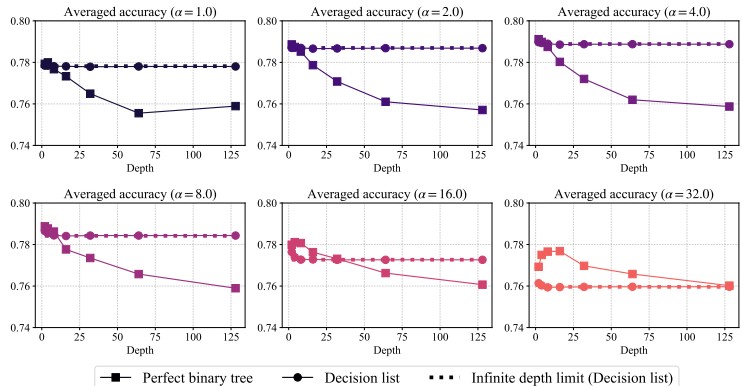

Figure 8: Averaged accuracy over 90 datasets. Horizontal dotted lines show the accuracy of decision lists with the infinite depth. The statistical significance is assessed in the Appendix.

The performance of decision lists already becomes almost consistent with their infinite depth limit when the depth reaches around 10. This suggests that we will no longer see significant changes in output for deeper decision lists. For small $\alpha$, asymmetric trees often perform better than symmetric trees, but the characteristics reverse for large $\alpha$.

**Computational complexity of the kernel.** Let $U = \sum_{d=1}^{D} \mathbb{1}_{\mathcal{Q}(d)>0}$, the number of depths connected to leaves. In general, the complexity for computing each kernel value for a pair of samples is $O(U)$. However, there are cases in which we can reduce the complexity to $O(1)$, such as in the case of an infinitely deep decision list as shown in Proposition 2, although $U = \infty$.

## 5    DISCUSSIONS

**Application to Neural Architecture Search (NAS).**  Arora et al. (2019) proposed using the NTK for *Neural Architecture Search* (NAS) (Elsken et al., 2019) for performance estimation. Such studies have been active in recent years (Chen et al., 2021; Xu et al., 2021; Mok et al., 2022). Our findings allow us to reduce the number of tree architecture candidates significantly. Theorem 3 tells us the existence of redundant architectures that do not need to be explored in NAS. The numerical experiments shown in Figure 8 suggest that we do not need to explore extremely deep tree structures even with asymmetric tree architecture.

**Analogy between decision lists and residual networks.**  Huang et al. (2020) showed that although the multi-layer perceptron without skip-connection (He et al., 2016) exhibits the degeneracy phenomenon, the multi-layer perceptron with skip-connection does not exhibit it. This is common to our situation, where skip-connection for the multi-layer perceptron corresponds to asymmetric structure for soft trees like decision lists. Moreover, Veit et al. (2016) proposed an interpretation of residual networks showing that they can be seen as a collection of many paths of differing lengths. This is similar to our case, because decision lists can be viewed as a collection of paths, i.e., rule sets, with different lengths. Therefore, our findings in this paper suggest that there may be a common reason why performance does not deteriorate easily as the depth increases.

## 6    CONCLUSIONS

We have introduced and studied the NTK induced by arbitrary tree architectures. Our theoretical analysis via the kernel provides new insights into the behavior of the infinite ensemble of soft trees: for different soft trees, if the number of leaves per depth is equal, the training behavior of their infinite ensembles in function space matches exactly, even if the tree architectures are not isomorphic. We have also shown, theoretically and empirically, that the deepening of asymmetric trees like decision lists does not necessarily induce the degeneracy phenomenon, although it occurs in symmetric perfect binary trees.

ACKNOWLEDGEMENT

This work was supported by JSPS, KAKENHI Grant Number JP21H03503, Japan and JST, CREST Grant Number JPMJCR22D3, Japan.

ETHICS STATEMENT

We believe that theoretical analysis of the NTK does not lead to harmful applications.

REPRODUCIBILITY STATEMENT

Proofs are provided in the Appendix. For numerical experiments and figures, reproducible source codes are shared in the supplementary material.

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

# A  PROOFS

## A.1  PROOF OF THEOREM 2

**Theorem 2.** *The NTK for an ensemble of $M$ soft rule sets with the depth $D$ converges in probability to the following deterministic kernel as $M \to \infty$,*

$$
\Theta^{(D,\text{Rule})}(\boldsymbol{x}_i, \boldsymbol{x}_j) \coloneqq \lim_{M \to \infty} \widehat{\Theta}_0^{(D,\text{Rule})}(\boldsymbol{x}_i, \boldsymbol{x}_j)
$$
$$
= \underbrace{D\, \Sigma(\boldsymbol{x}_i, \boldsymbol{x}_j)(\mathcal{T}(\boldsymbol{x}_i, \boldsymbol{x}_j))^{D-1}\dot{\mathcal{T}}(\boldsymbol{x}_i, \boldsymbol{x}_j)}_{\text{contribution from internal nodes}} + \underbrace{(\mathcal{T}(\boldsymbol{x}_i, \boldsymbol{x}_j))^D}_{\text{contribution from leaves}}\ .
$$

*Proof.* We consider the contribution from internal nodes $\Theta^{(D,\text{Rule,nodes})}$ and the contribution from leaves $\Theta^{(D,\text{Rule,leaves})}$ separately, such that

$$
\Theta^{(D,\text{Rule})}\,(\boldsymbol{x}_i, \boldsymbol{x}_j) = \Theta^{(D,\text{Rule,nodes})}\,(\boldsymbol{x}_i, \boldsymbol{x}_j) + \Theta^{(D,\text{Rule,leaves})}\,(\boldsymbol{x}_i, \boldsymbol{x}_j)\,. \tag{A.1}
$$

As for internal nodes, we have

$$
\frac{f^{(D,\text{Rule})}\,(\boldsymbol{x}_i, \boldsymbol{w}, \boldsymbol{\pi})}{\partial \boldsymbol{w}_{m,t}} = \frac{1}{\sqrt{M}}\boldsymbol{x}_i\dot{\sigma}(\boldsymbol{w}_{m,t}^\top \boldsymbol{x}_i)f_m^{(D-1,\text{Rule})}\,(\boldsymbol{x}_i, \boldsymbol{w}_{m,-t}, \boldsymbol{\pi}_m)\,, \tag{A.2}
$$

where we consider the derivative with respect to a node $t$, and $\boldsymbol{w}_{m,-t}$ denotes the internal node parameter matrix except for the parameters of the node $t$. Since there are $D$ possible locations for $t$, we obtain

$$
\Theta^{(D,\text{Rule,nodes})}\,(\boldsymbol{x}_i, \boldsymbol{x}_j) = D\, \Sigma(\boldsymbol{x}_i, \boldsymbol{x}_j)(\mathcal{T}(\boldsymbol{x}_i, \boldsymbol{x}_j))^{D-1}\dot{\mathcal{T}}(\boldsymbol{x}_i, \boldsymbol{x}_j), \tag{A.3}
$$

where

$$
\mathbb{E}_m\left[f_m^{(D,\text{Rule})}\,(\boldsymbol{x}_i, \boldsymbol{w}_m, \boldsymbol{\pi}_m)\,f_m^{(D,\text{Rule})}\,(\boldsymbol{x}_j, \boldsymbol{w}_m, \boldsymbol{\pi}_m)\right]
$$
$$
= \mathbb{E}_m\left[\underbrace{\sigma(\boldsymbol{w}_{m,1}^\top \boldsymbol{x}_i)\sigma(\boldsymbol{w}_{m,1}^\top \boldsymbol{x}_j)}_{\to \mathcal{T}(\boldsymbol{x}_i, \boldsymbol{x}_j)}\underbrace{\sigma(\boldsymbol{w}_{m,2}^\top \boldsymbol{x}_i)\sigma(\boldsymbol{w}_{m,2}^\top \boldsymbol{x}_j)}_{\to \mathcal{T}(\boldsymbol{x}_i, \boldsymbol{x}_j)}\cdots\underbrace{\sigma(\boldsymbol{w}_{m,D}^\top \boldsymbol{x}_i)\sigma(\boldsymbol{w}_{m,D}^\top \boldsymbol{x}_j)}_{\to \mathcal{T}(\boldsymbol{x}_i, \boldsymbol{x}_j)}\underbrace{\pi_{m,1}^2}_{\to 1}\right]
$$
$$
= (\mathcal{T}(\boldsymbol{x}_i, \boldsymbol{x}_j))^D \tag{A.4}
$$

is used. Here, the subscription "$\rightarrow$" means that the expected value of the corresponding term will be.

Similarly, for leaves,

$$\frac{f^{(D,\text{Rule})}(\boldsymbol{x}_i, \boldsymbol{w}, \boldsymbol{\pi})}{\partial \pi_{m,1}} = \frac{1}{\pi_{m,1}\sqrt{M}} f_m^{(D,\text{Rule})}(\boldsymbol{x}_i, \boldsymbol{w}_m, \boldsymbol{\pi}_m),\tag{A.5}$$

resulting in

$$\Theta^{(D,\text{Rule,leaves})}(\boldsymbol{x}_i, \boldsymbol{x}_j) = (\mathcal{T}(\boldsymbol{x}_i, \boldsymbol{x}_j))^D.\tag{A.6}$$

Combining Equation A.3 and Equation A.6, we obtain Equation 8. $\qquad\square$

## A.2 PROOF OF THEOREM 3

**Theorem 3.** *Let $\mathcal{Q} : \mathbb{N} \to \mathbb{N} \cup \{0\}$ be a function that receives any depth and returns the number of leaves connected to internal nodes at the input depth. For any tree architecture, the NTK for an ensemble of soft trees converges in probability to the following deterministic kernel as $M \to \infty$,*

$$\Theta^{(\text{ArbitraryTree})}(\boldsymbol{x}_i, \boldsymbol{x}_j) := \lim_{M \to \infty} \widehat{\Theta}_0^{(\text{ArbitraryTree})}(\boldsymbol{x}_i, \boldsymbol{x}_j) = \sum_{d=1}^{D} \mathcal{Q}(d)\, \Theta^{(d,\text{Rule})}(\boldsymbol{x}_i, \boldsymbol{x}_j).$$

*Proof.* We separate leaf and inner node contributions.

**Contribution from Internal Nodes.** For a soft Boolean operation, the following equations hold:

$$\mathbb{E}_m\left[(1 - \sigma(\boldsymbol{w}_{m,n}^\top \boldsymbol{x}_i))(1 - \sigma(\boldsymbol{w}_{m,n}^\top \boldsymbol{x}_j))\right] = \mathbb{E}_m\left[1 - \underbrace{\sigma(\boldsymbol{w}_{m,n}^\top \boldsymbol{x}_i)}_{\rightarrow 0.5} - \underbrace{\sigma(\boldsymbol{w}_{m,n}^\top \boldsymbol{x}_j)}_{\rightarrow 0.5}\right.$$
$$\left. + \sigma(\boldsymbol{w}_{m,n}^\top \boldsymbol{x}_i)\sigma(\boldsymbol{w}_{m,n}^\top \boldsymbol{x}_j)\right]$$
$$= \mathbb{E}_m[\sigma(\boldsymbol{w}_{m,n}^\top \boldsymbol{x}_i)\sigma(\boldsymbol{w}_{m,n}^\top \boldsymbol{x}_j)],\tag{A.7}$$

$$\mathbb{E}_m\left[\frac{\partial(1 - \sigma(\boldsymbol{w}_{m,n}^\top \boldsymbol{x}_i))}{\partial \boldsymbol{w}_{m,n}} \frac{\partial(1 - \sigma(\boldsymbol{w}_{m,n}^\top \boldsymbol{x}_j))}{\partial \boldsymbol{w}_{m,n}}\right] = \mathbb{E}_m\left[\boldsymbol{x}_i^\top \boldsymbol{x}_j \dot{\sigma}(\boldsymbol{w}_{m,n}^\top \boldsymbol{x}_i)\dot{\sigma}(\boldsymbol{w}_{m,n}^\top \boldsymbol{x}_j)\right]$$
$$= \mathbb{E}_m\left[\frac{\partial\sigma(\boldsymbol{w}_{m,n}^\top \boldsymbol{x}_i)}{\partial \boldsymbol{w}_{m,n}} \frac{\partial\sigma(\boldsymbol{w}_{m,n}^\top \boldsymbol{x}_j)}{\partial \boldsymbol{w}_{m,n}}\right].\tag{A.8}$$

Since each $\sigma(\boldsymbol{w}_{m,n}^\top \boldsymbol{x}_i)$ becomes $0.5$, although the term $1 - \sigma(\boldsymbol{w}_{m,n}^\top \boldsymbol{x}_i)$ is used instead of $\sigma(\boldsymbol{w}_{m,n}^\top \boldsymbol{x}_i)$ for the rightward flow in the tree, exactly the same limiting NTK can be obtained by treating $1 - \sigma(\boldsymbol{w}_{m,n}^\top \boldsymbol{x}_i)$ as $\sigma(\boldsymbol{w}_{m,n}^\top \boldsymbol{x}_i)$.

As for an inner node contribution, the derivative is obtained as

$$\frac{\partial f^{(\text{ArbitraryTree})}(\boldsymbol{x}_i, \boldsymbol{w}, \boldsymbol{\pi})}{\partial \boldsymbol{w}_{m,n}} = \frac{1}{\sqrt{M}} \sum_{\ell=1}^{\mathcal{L}} \pi_{m,\ell} \frac{\partial\mu_{m,\ell}(\boldsymbol{x}_i, \boldsymbol{w}_m)}{\partial \boldsymbol{w}_{m,n}}$$
$$= \frac{1}{\sqrt{M}} \sum_{\ell=1}^{\mathcal{L}} \pi_{m,\ell} S_{n,\ell}(\boldsymbol{x}_i, \boldsymbol{w}_m)\boldsymbol{x}_i \dot{\sigma}\left(\boldsymbol{w}_{m,n}^\top \boldsymbol{x}_i\right),\tag{A.9}$$

where

$$S_{n,\ell}(\boldsymbol{x}, \boldsymbol{w}_m) := \left(\prod_{n'=1}^{\mathcal{N}} \sigma\left(\boldsymbol{w}_{m,n'}^\top \boldsymbol{x}_i\right)^{\mathbb{1}_{(\ell \swarrow n')\&(n \neq n')}} \left(1 - \sigma\left(\boldsymbol{w}_{m,n'}^\top \boldsymbol{x}_i\right)\right)^{\mathbb{1}_{(n' \searrow \ell)\&(n \neq n')}}\right)(-1)^{\mathbb{1}_{n \searrow \ell}},$$
$$\tag{A.10}$$

and $\&$ is a logical conjunction. Since $\pi_{m,\ell}$ is initialized as zero-mean i.i.d. Gaussians with unit variances,

$$\mathbb{E}_m\left[\pi_{m,\ell}\pi_{m,\ell'}\right] = 0 \ \text{ if } \ \ell \neq \ell'. \tag{A.11}$$

Therefore, the inner node contribution for the limiting NTK is

$$\Theta^{(\text{ArbitraryTree,nodes})}(\boldsymbol{x}_i, \boldsymbol{x}_j)$$

$$= \mathbb{E}_m\left[\sum_{\ell=1}^{\mathcal{L}} \pi_{m,\ell}^2 S_{n,\ell}(\boldsymbol{x}_i, \boldsymbol{w}_m) S_{n,\ell}(\boldsymbol{x}_j, \boldsymbol{w}_m)\boldsymbol{x}_i^\top \boldsymbol{x}_j \dot{\sigma}\left(\boldsymbol{w}_{m,n}^\top \boldsymbol{x}_i\right)\dot{\sigma}\left(\boldsymbol{w}_{m,n}^\top \boldsymbol{x}_j\right)\right]$$

$$= \Sigma(\boldsymbol{x}_i, \boldsymbol{x}_j)\dot{\mathcal{T}}(\boldsymbol{x}_i, \boldsymbol{x}_j)\mathbb{E}_m\left[\sum_{\ell=1}^{\mathcal{L}} S_{n,\ell}(\boldsymbol{x}_i, \boldsymbol{w}_m)S_{n,\ell}(\boldsymbol{x}_j, \boldsymbol{w}_m)\right]. \tag{A.12}$$

Suppose leaf $\ell$ is connected to an internal node of depth $d$. With Equation A.7 and Equation A.8, we obtain

$$\mathbb{E}_m\left[S_{n,\ell}(\boldsymbol{x}_i, \boldsymbol{w}_m)S_{n,\ell}(\boldsymbol{x}_j, \boldsymbol{w}_m)\right] = (\mathcal{T}(\boldsymbol{x}_i, \boldsymbol{x}_j))^{d-1}. \tag{A.13}$$

Therefore, considering all leaves,

$$\Theta^{(\text{ArbitraryTree,nodes})}(\boldsymbol{x}_i, \boldsymbol{x}_j) = \sum_{d=1}^{D} \mathcal{Q}(d)\Theta^{(d,\text{Rule,nodes})}, \tag{A.14}$$

where $\Theta^{(d,\text{Rule,nodes})}$ is introduced in Equation A.3

**Contribution from Leaves.** As for the contribution from leaves, the derivative is obtained as

$$\frac{\partial f^{(\text{ArbitraryTree})}(\boldsymbol{x}_i, \boldsymbol{w}, \boldsymbol{\pi})}{\partial \pi_{m,\ell}} = \frac{1}{\sqrt{M}}\mu_{m,\ell}(\boldsymbol{x}_i, \boldsymbol{w}_m). \tag{A.15}$$

Since $\boldsymbol{w}_{m,n}$ used in $\mu_{m,\ell}(\boldsymbol{x}_i, \boldsymbol{w}_m)$ is initialized as zero-mean i.i.d. Gaussians, contribution from leaves on the limiting NTK induced by arbitrary tree architecture is:

$$\Theta^{(\text{ArbitraryTree,leaves})}(\boldsymbol{x}_i, \boldsymbol{x}_j) = \mathbb{E}_m\left[\sum_{\ell=1}^{\mathcal{L}} \mu_{m,\ell}(\boldsymbol{x}_i, \boldsymbol{w}_m)\mu_{m,\ell}(\boldsymbol{x}_j, \boldsymbol{w}_m)\right]$$

$$= \sum_{d=1}^{D} \mathcal{Q}(d)\Theta^{(d,\text{Rule,leaves})}, \tag{A.16}$$

where $\Theta^{(d,\text{Rule,leaves})}$ is introduced in Equation A.6 $\hfill\square$

### A.3 PROOF OF THEOREM 4

**Theorem 4.** *Let $\lambda_{\min}$ and $\lambda_{\max}$ be the minimum and maximum eigenvalues of the limiting NTK. Assume $\|\boldsymbol{x}_i\|_2 = 1$ for all $i \in [N]$ and $\boldsymbol{x}_i \neq \boldsymbol{x}_j$ ($i \neq j$). For ensembles of arbitrary soft trees with the NTK initialization trained under gradient flow with a learning rate $\eta < 2/(\lambda_{\min} + \lambda_{\max})$ and a positive finite scaling factor $\alpha$, we have, with high probability,*

$$\sup\left|\widehat{\Theta}_\tau^{(\text{ArbitraryTree})}(\boldsymbol{x}_i, \boldsymbol{x}_j) - \widehat{\Theta}_0^{(\text{ArbitraryTree})}(\boldsymbol{x}_i, \boldsymbol{x}_j)\right| = \mathcal{O}\left(\frac{1}{\sqrt{M}}\right). \tag{A.17}$$

*Proof.* To prove that the kernel does not move during training, we need to show the positive definiteness of the kernel and the local Lipschitzness of the model Jacobian at initialization $\boldsymbol{J}(\boldsymbol{x}, \boldsymbol{\theta})$, whose $(i, j)$ entry is $\frac{\partial f(\boldsymbol{x}_i, \boldsymbol{\theta})}{\partial \theta_j}$ where $\theta_j$ is a $j$-th component of $\boldsymbol{\theta}$:

**Lemma 1** (Lee et al. (2019)). *Assume that the limiting NTK induced by any model architecture is positive definite for input sets $\boldsymbol{x}$, such that minimum eigenvalue of the NTK $\lambda_{\min} > 0$. For models with local Lipschitz Jacobian trained under gradient flow with a learning rate $\eta < 2(\lambda_{\min} + \lambda_{\max})$, we have with high probability:*

$$\sup\left|\widehat{\Theta}_\tau^*(\boldsymbol{x}_i, \boldsymbol{x}_j) - \widehat{\Theta}_0^*(\boldsymbol{x}_i, \boldsymbol{x}_j)\right| = \mathcal{O}\left(\frac{1}{\sqrt{M}}\right). \tag{A.18}$$

The local Lipschitzness of the soft tree ensemble's Jacobian at initialization is already proven, even with arbitrary tree architectures:

**Lemma 2** (Kanoh & Sugiyama (2022)). *For soft tree ensemble models with the NTK initialization and a positive finite scaling factor $\alpha$, there is $K > 0$ such that for every $C > 0$, with high probability, the following holds:*

$$\begin{cases} \|\boldsymbol{J}(\boldsymbol{x}, \boldsymbol{\theta})\|_F & \leq K \\ \|\boldsymbol{J}(\boldsymbol{x}, \boldsymbol{\theta}) - \boldsymbol{J}(\boldsymbol{x}, \tilde{\boldsymbol{\theta}})\|_F & \leq K\|\boldsymbol{\theta} - \tilde{\boldsymbol{\theta}}\|_2 \end{cases}, \forall \boldsymbol{\theta}, \tilde{\boldsymbol{\theta}} \in B\left(\boldsymbol{\theta}_0, C\right), \tag{A.19}$$

*where*

$$B\left(\theta_0, C\right) := \{\boldsymbol{\theta} : \|\boldsymbol{\theta} - \boldsymbol{\theta}_0\|_2 < C\}. \tag{A.20}$$

As for the positive definiteness, $\Theta^{(D,\text{PB})}(\boldsymbol{x}_i, \boldsymbol{x}_j)$ is known to be positive definite.

**Lemma 3** (Kanoh & Sugiyama (2022)). *For infinitely many perfect binary soft trees with any depth and the NTK initialization, the limiting NTK is positive definite if $\|\boldsymbol{x}_i\|_2 = 1$ for all $i \in [N]$ and $\boldsymbol{x}_i \neq \boldsymbol{x}_j$ $(i \neq j)$.*

Since $\Theta^{(D,\text{PB})}(\boldsymbol{x}_i, \boldsymbol{x}_j)$ is equivalent to $\Theta^{(D,\text{Rule})}(\boldsymbol{x}_i, \boldsymbol{x}_j)$ up to constant multiple, $\Theta^{(D,\text{Rule})}(\boldsymbol{x}_i, \boldsymbol{x}_j)$ is positive definite under the same assumption. Besides, since $\Theta^{(\text{ArbitraryTree})}(\boldsymbol{x}_i, \boldsymbol{x}_j)$ is represented by the summation of $\Theta^{(D,\text{Rule})}(\boldsymbol{x}_i, \boldsymbol{x}_j)$ as in Theorem 3, $\Theta^{(\text{ArbitraryTree})}(\boldsymbol{x}_i, \boldsymbol{x}_j)$ is also positive definite under the same assumption.

These results show that the limiting NTK induced by arbitrary tree architecture also does not change during training. $\qquad\square$

# B  DETAILS OF NUMERICAL EXPERIMENTS

## B.1  DATASET ACQUISITION

We used the UCI datasets (Dua & Graff, 2017) preprocessed by Fernández-Delgado et al. (2014)[1]. We selected 90 datasets with less than 5000 data points as in Arora et al. (2020) and Kanoh & Sugiyama (2022).

## B.2  MODEL SPECIFICATIONS

We used scikit-learn[2] to perform kernel regression. The regularization strength is set to be a tiny value ($1.0 \times 10^{-8}$) so that it becomes almost ridge-less regression.

## B.3  COMPUTATIONAL RESOURCE

We ran all experiments on 2.20 GHz Intel Xeon E5-2698 CPU and 252 GB of memory with Ubuntu Linux (version: 4.15.0-117-generic).

## B.4  STATISTICAL SIGNIFICANCE

We conducted a Wilcoxon signed rank test for 90 datasets to check the statistical significance of the differences between performances on a perfect binary tree and a decision list. Figure A.1 shows the p-values. Statistically significant differences can be observed for areas where the differences appear large in Figure 8, such as when $\alpha$ is small and $D$ is large. We used Bonferroni correction to account for multiple testing, and the resulting significance level of the p-value is about $0.0012$ for 5 percent confidence level. An asterisk "*" is placed in Figure A.1 with statistically significant result even

---

[1] http://persoal.citius.usc.es/manuel.fernandez.delgado/papers/jmlr/data.tar.gz

[2] https://scikit-learn.org/stable/modules/generated/sklearn.kernel_ridge.KernelRidge.html

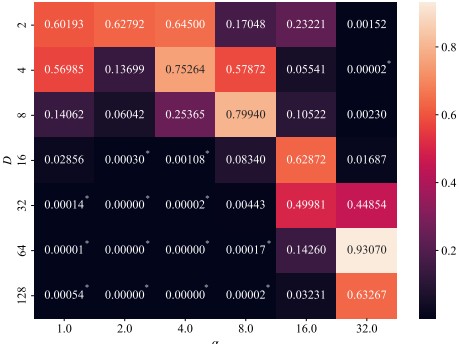

Figure A.1: P-values of the Wilcoxon signed rank test for results on perfect binary trees and decision lists with different parameters.

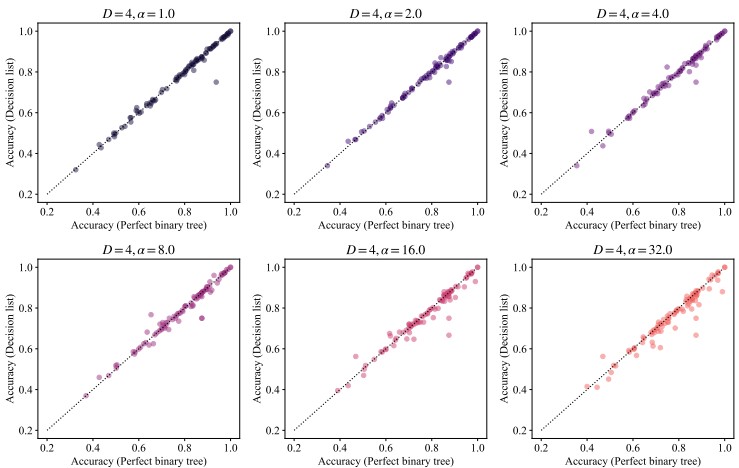

Figure A.2: Performance comparisons between the kernel regression with the limiting NTK induced by the perfect binary tree and the decision list on the UCI datasets with $D = 4$.

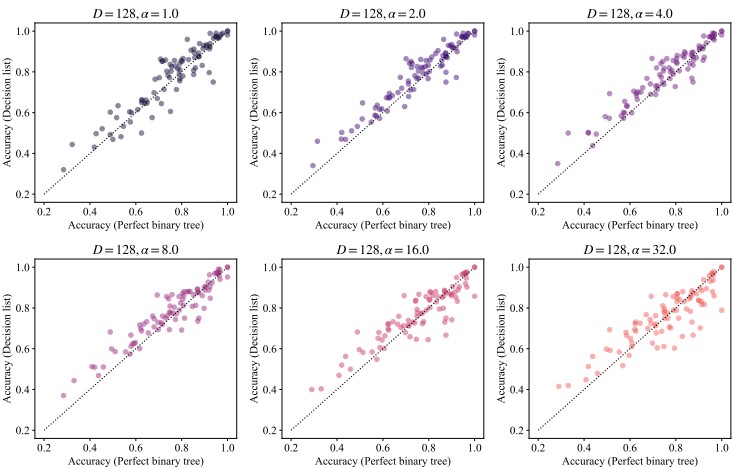

Figure A.3: Performance comparisons between the kernel regression with the limiting NTK induced by the perfect binary tree and the decision list on the UCI datasets with $D = 128$.

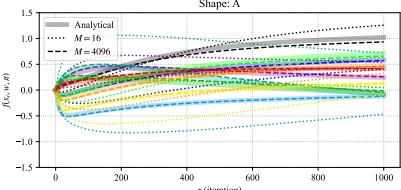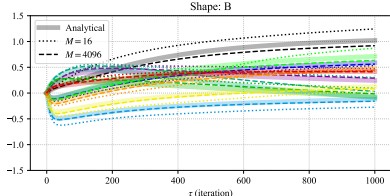

Figure A.4: Output dynamics for test data points. Each line color corresponds to each data point. Analytical trajectories are the same for both shapes A and B.

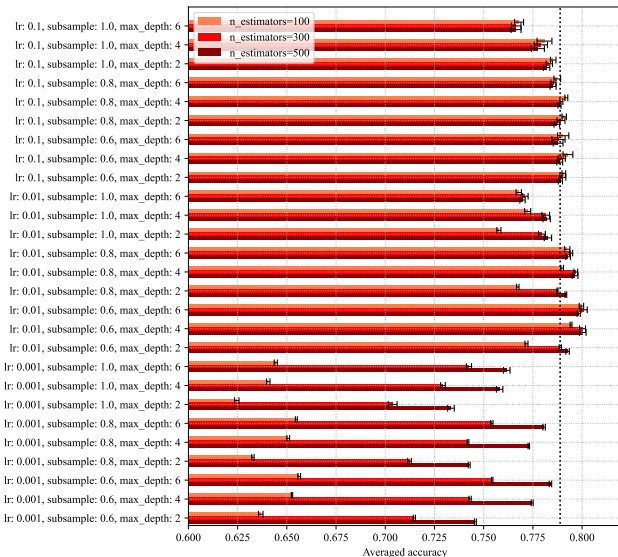

Figure A.5: Averaged gradient boosting decision tree accuracy over 90 datasets. The random seeds are changed five times and their averaged performance is shown. Their maximum and minimum performance is shown by the error bars. The vertical dotted line shows the performance of the infinitely deep decision list with $\alpha = 4.0$.

after correction. For cases where the symmetry of the tree does not produce a large difference, such as at the depth of 2, the difference in performance is often not statistically significant.

Figures A.2 and A.3 show scatter-plots between the performance of the kernel regression with the limiting NTK induced by the perfect binary tree and the decision list. The deeper the tree, the larger the difference between them.

## C  ADDITIONAL NUMERICAL EXPERIMENTS

### C.1  COMPARISON TO THE ENSEMBLES OF FINITE SOFT TREES

Figure A.4 illustrates output trajectories for two different tree architectures shown in Figure 3 during gradient descent training with analytical trajectories obtained from the limiting kernel: $f(v, \theta_\tau) = H(v, x)H(x, x)^{-1}(I - \exp[-\eta H(x, x)\tau])y$ (Lee et al., 2019), where $v \in \mathbb{R}^F$ is an arbitrary input and $x \in \mathbb{R}^{F \times N}$ and $y \in \mathbb{R}^N$ are the training dataset and targets, respectively. The setup is the same with Figure 4. Since the limiting kernel is the same for both architectures, analytical trajectories are the same for both left and right panels. As the number of trees increases, we can see that the behavior of finite trees approaches the analytical trajectory obtained by the NTK.

## C.2 COMPARISON TO THE GRADIENT BOOSTING DECISION TREE

For reference information, we show experimental results for the gradient boosting decision tree. The experimental procedure is the same as that in Section 4.3. We used scikit-learn[3] for the implementation. As for hyperparameters, we used `max_depth` in $\{2, 4, 6\}$, `subsample` in $\{0.6, 0.8, 1.0\}$, `learning_rate` in $\{0.1, 0.01, 0.001\}$, and `n_estimators` (the number of trees) in $\{100, 300, 500\}$. Other parameters were set to be default values of the library.

Figure A.5 shows the averaged accuracy over 90 datasets. We used five random seeds $\{0, 1, 2, 3, 4\}$ and their mean, minimum, and maximum performances are reported. When we use the best parameter, its averaged accuracy is $0.8010$, which is slightly better than the performance of infinitely deep decision list: $0.7889$ with $\alpha = 4.0$ as shown in the dotted line in Figure A.5. When we look at each dataset, however, the infinitely deep decision list is superior to the gradient boosting decision tree for 35 out of 90 datasets. One is not necessarily better than the other, and their inductive biases may or may not be appropriate for each dataset.

---

[3]`https://scikit-learn.org/stable/modules/generated/sklearn.ensemble.GradientBoostingRegressor.html`

