# OpenReview forum: "Analyzing Tree Architectures in Ensembles via Neural Tangent Kernel"
_ICLR.cc/2023/Conference — ICLR 2023 poster_

### Official Review · Reviewer_ck6w · 2022-10-24

**Confidence:** 4
**Correctness:** 4
**Technical Novelty And Significance:** 4
**Empirical Novelty And Significance:** 4
**Recommendation:** 8

**Clarity, Quality, Novelty And Reproducibility:**

In my opinion, the paper is very well-written. In particular, section 2 is very valuable and provides a good introduction to the problem and the tools used. However, I think that the comments on Figure 1 should be extended and clarified. The very short paragraph on the computational complexity of the kernel should also be improved.

The results are new and extend a very recent paper published in ICLR 2022. The numerical experiments are in accordance with the theoretical analysis.

If we have to find a flaw, I think the subject is quite specific since it deals with infinite soft tree ensembles in the NTK framework. However, recent literature shows that ensemble learning is a major topic.

Typos:
- the structure is symmetry (page 1)
- when the number $M$ of tress gets larger (page 7)

**Strength And Weaknesses:**

This paper extends in a non-trivial way the understanding of infinite soft tree ensembles that was previously limited to binary trees. In particular, it is shown that the tree topology has an effect only through the number of leaves at each depth. The conclusion on the degeneracy effect observed for binary trees but not for decision list is also highly informative. Finally, the numerical results support the theoretical findings.

I have the two following questions only:
- If I understood correctly, all the soft trees in Theorem 3 share the same topology. Would it not be relevant to study the case where not all trees have the same topology? I think this could encompass some interesting practical cases.
- Theorem 3 generalizes Theorem 1 of [Kanoh and Sugiyama, 2022] established for binary trees to any tree architecture. The idea of the proof is to see trees as combinations of independent rule sets, which is quite different from the proof developed in [Kanoh and Sugiyama, 2022]. Can the authors comment on the differences between the two proofs, both their advantages and drawbacks?


**Summary Of The Paper:**

This paper deals with ensemble learning where the weak learners are soft trees. More precisely, it aims to study from a theoretical perspective soft tree ensembles when the number of weak learners goes to infinity. To this end, the authors investigate the NTK (Neural Tangent Kernel) induced by them and prove several non-trivial results, which seems to me of great interest, without any assumption on the topology of the soft trees. They are in addition validated from numerical experiments. The paper is an extension of an article published in ICLR 2022 by Kanoh and Sugiyama hereafter referred to as [Kanoh and Sugiyama, 2022].

**Summary Of The Review:**

My opinion on this paper is very positive: it is well written and organized, improves recent results, and the theoretical and numerical results are non-trivial and in agreement with each other.

---

> ### Author Response · Authors · 2022-11-17
> **Authors’ Response to Reviewer ck6w**
>
> Thank you for your review.
>
> ----
>
> > If I understood correctly, all the soft trees in Theorem 3 share the same topology. Would it not be relevant to study the case where not all trees have the same topology? I think this could encompass some interesting practical cases.
>
> (Same reply to Reviewer AJRY)
>
> It is actually possible to theoretically analyze ensembles with various tree architectures mixed together. Assuming the existence of an infinite number of trees, the NTK can be computed analytically if the amount (ratio) of each structure in the ensemble is known.
> For example, if perfect binary trees of depth 2 and perfect binary trees of depth 3 exist in the ratio 1:1, the resulting kernel is $\Theta^{(2, \text{PB})}(\boldsymbol{x}_i, \boldsymbol{x}_j) + \Theta^{(3, \text{PB})}(\boldsymbol{x}_i, \boldsymbol{x}_j)$. This property holds even if the tree architecture is arbitrary.
>
> [Explanation]
>
> Assume that the function that we want to calculate the NTK contains two types of trees.
>
> $f(\boldsymbol{x}_i, \boldsymbol{\theta}) = g(\boldsymbol{x}_i, \boldsymbol{\theta}_1) + h(\boldsymbol{x}_i, \boldsymbol{\theta}_2)$, where parameters $\boldsymbol{\theta}_1 \in \mathbb{R}^{N_1}$, $\boldsymbol{\theta}_2 \in \mathbb{R}^{N_2}$, and $\boldsymbol{\theta} \in \mathbb{R}^{N_1+N_2}$ is a concatenated vector.
>
> For example, the functions $g$ and $h$ represent tree ensemble models with a depth of 2 and 3, respectively. Let us assume that both $g$ and $h$ contain the same number of trees and, for simplicity, suppose the case where they exist in a 1:1 ratio. Note that it is easy to generalize it for ensembles containing various tree architectures.
>
> The NTK induced by this model can be decomposed into the sum of the NTKs of each tree architecture as follows:
> $\left\langle \frac{\partial f(\boldsymbol{x}_i, \boldsymbol{\theta})}{\partial \boldsymbol{\theta}},\frac{\partial f(\boldsymbol{x}_j, \boldsymbol{\theta})}{\partial \boldsymbol{\theta}} \right\rangle = \left\langle \frac{\partial g(\boldsymbol{x}_i, \boldsymbol{\theta}_1)}{\partial \boldsymbol{\theta}_1},\frac{\partial g(\boldsymbol{x}_j, \boldsymbol{\theta}_1)}{\partial \boldsymbol{\theta}_1} \right\rangle + \left\langle \frac{\partial h(\boldsymbol{x}_i, \boldsymbol{\theta}_2)}{\partial \boldsymbol{\theta}_2},\frac{\partial h(\boldsymbol{x}_j, \boldsymbol{\theta}_2)}{\partial \boldsymbol{\theta}_2} \right\rangle$
>
> ----
>
> > Theorem 3 generalizes Theorem 1 of [Kanoh and Sugiyama, 2022] established for binary trees to any tree architecture. The idea of the proof is to see trees as combinations of independent rule sets, which is quite different from the proof developed in [Kanoh and Sugiyama, 2022]. Can the authors comment on the differences between the two proofs, both their advantages and drawbacks?
>
> Our proof technique is more general yet simpler than that in [1].
> By incorporating properties such as the symmetric properties of the decision function and the statistical independence of the leaf parameters, we successfully extend the proof to treat arbitrary tree architectures. It is possible to prove Theorem 1 of [1] using our proof technique.
>
> ----
>
> > Typos
>
> We have fixed them in the revised paper. Thank you for pointing that out.
>
> ----
>
> [1] Kanoh&Sugiyama (2022), A Neural Tangent Kernel Perspective of Infinite Tree Ensembles, ICLR2022

---

### Official Review · Reviewer_RPPM · 2022-10-24

**Confidence:** 4
**Correctness:** 3
**Technical Novelty And Significance:** 3
**Empirical Novelty And Significance:** 3
**Recommendation:** 6

**Clarity, Quality, Novelty And Reproducibility:**

The theoretical results extend work of R Kanoh et al (ICLR 2022) to the case of non-perfect binary trees, so it is limited novelty.
The overall clarity of the work is high.
The work seems to be reproducible as the code is provided.

**Strength And Weaknesses:**

Strength:
-- rigorous theoretical analysis
-- comparison of different architectures
-- discussion of connection with residual networks

Weaknesses:
-- experimental validation is mostly absent, since authors claim that kernel is non degenerate seeing how stochastic gradient optimization is able to overfit on arbitrary function and it's generalisation gaps (especially compared to classical gradient boosting) are desired to support theoretical claims in the situation of limited novelty.
-- experiments that are present are done on top of Kernel Regression with kernel defined by NTK, it is interesting to see how stochastic gradient descent results are aligned with Kernel Regression results as in practical setting applications of Kernel Regression is limited due to necessity to invert kernel matrix.
-- No link with classical gradient boosting

**Summary Of The Paper:**

The paper analyses tree architectures for neural trees ensembles via NTK. Authors provide important insights on how the structure of trees affects NTK and consider depth limit of NTK. Experimental results are based on kernel regression with arising NTK.

**Summary Of The Review:**

Theoretical results are novel for literature but the novelty is limited by the fact that results slightly extend analysis of R Kanoh et al (ICLR 2022). Paper has rigorous theoretical analysis but lacks more detailed experimentations. Nevertheless, results are interesting and worth presenting as provide interesting insight on the kernel structure.

---

> ### Author Response · Authors · 2022-11-17
> **Authors’ Response to Reviewer RPPM**
>
> Thank you for your review.
>
> ----
>
> > experimental validation is mostly absent, since authors claim that kernel is non degenerate seeing how stochastic gradient optimization is able to overfit on arbitrary function and it's generalisation gaps (especially compared to classical gradient boosting) are desired to support theoretical claims in the situation of limited novelty.
>
> We agree that a comparison with gradient boosting may be a good reference. Therefore, we have added the result to Appendix C.2 and Figure A.5 in the revised paper.
>
> Nevertheless, we argue that comparison with gradient boosting is not essential as it does not support our theoretical claims. As we have discussed degeneracy in Sections 4.2 and 4.3, in decision lists, degeneracy does not occur even when trees are very deep, and we have already verified this in our numerical experiments as shown in Figure 8.
>
>
> ----
>
> > experiments that are present are done on top of Kernel Regression with kernel defined by NTK, it is interesting to see how stochastic gradient descent results are aligned with Kernel Regression results as in practical setting applications of Kernel Regression is limited due to necessity to invert kernel matrix.
>
> We have added results of ensembles of finite soft trees in Appendix C.1 and Figure A.4 in the revised paper. We can see from this numerical experiment that, as the number of trees increases, the behavior of finite trees approaches the analytical trajectory obtained by the NTK.
>
> About the advantage of kernel methods, it is known that the kernel method using the NTK is known to perform well when data is small [1], and in some cases, it can be trained even faster than training an equivalent model using GPUs [2]. Therefore kernel methods have their own practical applications.
>
> ----
>
> > No link with classical gradient boosting
>
> Although gradient boosting is also based on decision trees, it is a coordinate descent method in function space, hence it is currently difficult to analyze directly using the NTK, as the NTK is based on gradient descent in function space. We agree that this viewpoint is still important and would like to consider it as a future research topic.
>
> ----
>
> [1] Arora et al. (2020), Harnessing the Power of Infinitely Wide Deep Nets on Small-data Tasks, ICLR2020
>
> [2] Du et al (2019), Fusing Graph Neural Networks with Graph Kernels, NeurIPS2019

---

> > ### Comment · Reviewer_RPPM · 2022-12-12
> > **Thx**
> >
> > Thank you for your response and clarifications, overall I’m satisfied

---

### Official Review · Reviewer_gydX · 2022-10-25

**Confidence:** 3
**Correctness:** 4
**Technical Novelty And Significance:** 4
**Empirical Novelty And Significance:** 2
**Recommendation:** 5

**Clarity, Quality, Novelty And Reproducibility:**

This paper is well-written but it would be better if more concrete results of TNTK (e.g., usages, generalization or optimization) are given. The novelty is incremental.

**Strength And Weaknesses:**

Strength:

This paper generalizes the previous result on the perfect binary tree to the arbitrary ones. This result gives us that only the number of leaves at each depth affects the TNTK. As the authors suggested, this is useful for the neural architect search (NAS) where the candidate models are infinite ensemble of soft trees.


Weakness:

Although the results of this paper are concrete, a justification of the TNTK is weak and it is still unclear which tasks/applications require the TNTK. The contribution would be much stronger if it shows some practical applications, e.g., Neural Architecture Search (NAS).

A performance gap between the perfect binary tree and the decision list comes from the depth degeneracy. The authors give a simple intuition on the behavior of region split, but this intuition is not sufficient to explain the performance gap. It would be better if more rigorous reasoning is provided.

The performance of the TNTK in section 4.3 depends on the choice of scaling factor ``alpha’’. Does this alpha control degeneracy of the TNTK? If so, it would be great to compare the degeneracy to practical performance (with different choices of alpha). And it would be better to compare the TNTK with other (sigmoid-like) activations.


**Summary Of The Paper:**

This paper studies the TNTK, i.e., neural tangent kernel induced by soft tree ensembles. The authors first provide a closed-form expression of the TNTK with arbitrary architecture. This is a generalization of the previous work (Kanoh and Sugiyama, 2022), which only studies the perfect binary tree. The underlying idea is to decompose a tree structure to rule sets, and to utilize their superposition property. Hence, the TNTK is the weighted summation of that of each rule set. This also implies that only the number of leaves at each depth affects the TNTK. Hence, different tree structures can have identical training behaviors and generalization performance under the infinite ensemble limit. Second, they deeply analyze a corner case, i.e., a tree growing in only one direction called a decision list. In particular, the TNTK of the decision tree does not degenerate while that of the perfect binary tree does. Such non-degeneracy property makes the performance of the TNTK not decrease when the depth becomes large. Experimental results also well support their result.


**Summary Of The Review:**

This paper well generalizes the previous results on the perfect binary tree, but the novelty is weak, and more concrete justification on the TNTK would be needed.

---

> ### Author Response · Authors · 2022-11-17
> **Authors’ Response to Reviewer gydX**
>
> Thank you for your review.
>
> ----
>
> > Although the results of this paper are concrete, a justification of the TNTK is weak and it is still unclear which tasks/applications require the TNTK. The contribution would be much stronger if it shows some practical applications, e.g., Neural Architecture Search (NAS).
>
> It has been already widely known that the NTK is a powerful tool for the theoretical analysis of machine learning models [1, 2, 3]. Since soft trees have been actively used in various studies and applications in recent years, we believe that our contribution, the NTK for soft trees with arbitrary architectures (TNTK), will be an important milestone for the theoretical analysis of their behavior. Moreover, the overparameterization of machine learning models has been receiving more and more attention in recent years, and the NTK also contributes to understanding the behavior of overparameterized models.
>
> In addition to being a tool for theoretical understanding, the TNTK can also be beneficial in practical applications.
> As [4] shows, kernel methods using the NTK are known to be able to perform very well when the size of data is relatively small, which is one of the practical applications of the NTK. [5] has already compared the performance of infinite ensembles of perfect binary soft trees obtained via the TNTK with that of infinite-width MLPs, and it has empirically demonstrated the computational efficiency and the good empirical prediction performance on some datasets. This result already shows the effectiveness of the TNTK, and we have successfully extended the applicability of the TNTK using the entirely new proof technique in our submission.
> As you pointed out and we have described in the paper, neural architecture search is definitely a relevant application for the (T)NTK, while it is not in the scope of our paper and can be an interesting future research topic.
>
> ----
>
> > A performance gap between the perfect binary tree and the decision list comes from the depth degeneracy. The authors give a simple intuition on the behavior of region split, but this intuition is not sufficient to explain the performance gap. It would be better if more rigorous reasoning is provided.
>
> As we have already discussed in Section 4.2 and Figure 7, the depth degeneracy leads to a kernel whose behavior is extreme; that is, it gives a very large value for a pair of vectors only if they are fairly similar with each other and gives an almost zero value if not. As we can imagine, such an extreme kernel usually does not work well for real-world datasets, while there could be exceptional cases as we have shown in Figure A.3.
>
> ----
>
> > The performance of the TNTK in section 4.3 depends on the choice of scaling factor ``alpha’’. Does this alpha control degeneracy of the TNTK? If so, it would be great to compare the degeneracy to practical performance (with different choices of alpha). And it would be better to compare the TNTK with other (sigmoid-like) activations.
>
> We do not think the alpha, a parameter used to characterize the shape of the decision function, controls degeneracy. Instead, the degeneracy is caused by the tree depth in the case of the complete binary tree. For example, when the depth of the perfect binary tree is infinite, all splitting regions become infinitely small, hence every data point falls into a unique leaf. In Figure 8, we can see that the performance (averaged accuracy) does not always drop if alpha increases by side-by-side comparison between plots of different alpha, which supports our claim.
>
> Closed-form kernels with sigmoid-like activation functions that are not based on Erf, such as SparseMax, are not known. If the activation function is based on Erf, a closed-form kernel like Equations (6) and (7) can be obtained [6, 7]. Therefore, in our paper, we try to check the effect of changes in the activation function by varying alpha values.
>
> ----
>
> [1] Jacot et al (2018), Neural Tangent Kernel: Convergence and Generalization in Neural Networks, NeurIPS2018
>
> [2] Lee et al (2019), Wide Neural Networks of Any Depth Evolve as Linear Models Under Gradient Descent, NeurIPS2019
>
> [3] Alemohammad et al (2021), The Recurrent Neural Tangent Kernel, ICLR2021
>
> [4] Arora et al. (2020), Harnessing the Power of Infinitely Wide Deep Nets on Small-data Tasks, ICLR2020
>
> [5] Kanoh&Sugiyama (2022), A Neural Tangent Kernel Perspective of Infinite Tree Ensembles, ICLR2022
>
> [6] Williams (1996), Computing with Infinite Networks, NIPS1996
>
> [7] Lee et al. (2019), ​​Wide Neural Networks of Any Depth Evolve as Linear Models Under Gradient Descent, NeurIPS2019

---

### Official Review · Reviewer_AJRY · 2022-10-26

**Confidence:** 2
**Correctness:** 3
**Technical Novelty And Significance:** 4
**Empirical Novelty And Significance:** 4
**Recommendation:** 6

**Clarity, Quality, Novelty And Reproducibility:**

In general, this paper is of good quality with high novelty and I really enjoy reading this paper.

**Strength And Weaknesses:**

Strengths:
1. This paper extends the theoretical study of (Kanoh & Sugiyama, 2022) on the perfect binary trees to many other types of soft trees in the literature, which is important to provide a more practical theoretical understanding in this field.
2. The theoretical results and insights of this paper are quite interesting and usually have been validated using empirical experiments. These results also provide certain implications for the application of these tree ensembles, which can be helpful in practice.
3. The whole paper is well-written and easy to follow.

weaknesses:
1. If I understand correctly, the theoretical analysis of this paper is based on the ensemble of the same type of soft trees with the same depth. So, will the theoretical results in this paper be also applicable to the ensemble of the different types of soft trees with various depths (even only empirical verification will be enough)? Because such an ensemble seems to be common in practice.
2. More interpretation for theorem 3 (e.g., how "theorem 3 tells us that the limiting NTK depends on only the number of leaves at each depth with respect to tree architecture" specifically) can make the importance of theorem 3 clearer as theorem 3 is the first and also the most fundamental contribution of this paper.
3. For the interpretation of Figure 3, it's unclear why NTK can be connected with the training behavior and generalization performance of tree ensembles. Any theoretical support for this, like the ones in (Jacot et al., 2018) and [1,2], but with respect to the soft tree ensemble models? I think more explanations need to be provided here since it's an important insight drawn from theorem 3, which also serves as the second contribution shown in the introduction.
4. Mentioning "ensemble" in the caption of figure 3 can make this result easier to be understood. There are many lines (w/ different colors) without labels in figures 4 and 6. I hope the authors can further polish these results to make them more readable.

[1] Arora, Sanjeev, Simon S. Du, Wei Hu, Zhiyuan Li, and Ruosong Wang. 2019. “Fine-Grained Analysis of Optimization and Generalization for Overparameterized Two-Layer Neural Networks.”
[2] Cao, Yuan, and Quanquan Gu. 2019. “Generalization Bounds of Stochastic Gradient Descent for Wide and Deep Neural Networks.”

**Summary Of The Paper:**

While existing work (Kanoh & Sugiyama, 2022) has introduced NTK to analyze the ensemble behavior of an infinite number of soft trees, such a study has mainly targeted the perfect binary tree, leaving the theoretical understanding of other types of widely used soft trees (e.g., decision list, rule set) unknown yet. To this end, this paper follows this line of work to provide theoretical analyses for other types of widely used soft trees via NTK and provides many interesting insights.

**Summary Of The Review:**

Overall, I am quite interested in this work and excited to see that NTK has inspired the theoretical analysis in other fields, e.g., the soft tree ensembles studied in this paper. However, as I am not professional in this area, currently I can only deliver a score of 6 for this work with small confidence. But I believe that this paper is already of good quality.

---

> ### Author Response · Authors · 2022-11-17
> **Authors’ Response to Reviewer AJRY**
>
> Thank you for your review.
>
> ----
>
> > the theoretical analysis of this paper is based on the ensemble of the same type of soft trees with the same depth. So, will the theoretical results in this paper be also applicable to the ensemble of the different types of soft trees with various depths (even only empirical verification will be enough)?
>
> (Same reply to Reviewer ck6w)
>
> It is actually possible to theoretically analyze ensembles with various tree architectures mixed together. Assuming the existence of an infinite number of trees, the NTK can be computed analytically if the amount (ratio) of each structure in the ensemble is known.
> For example, if perfect binary trees of depth 2 and perfect binary trees of depth 3 exist in the ratio 1:1, the resulting kernel is $\Theta^{(2, \text{PB})}(\boldsymbol{x}_i, \boldsymbol{x}_j) + \Theta^{(3, \text{PB})}(\boldsymbol{x}_i, \boldsymbol{x}_j)$. This property holds even if the tree architecture is arbitrary.
>
> [Explanation]
>
> Assume that the function that we want to calculate the NTK contains two types of trees.
>
> $f(\boldsymbol{x}_i, \boldsymbol{\theta}) = g(\boldsymbol{x}_i, \boldsymbol{\theta}_1) + h(\boldsymbol{x}_i, \boldsymbol{\theta}_2)$, where parameters $\boldsymbol{\theta}_1 \in \mathbb{R}^{N_1}$, $\boldsymbol{\theta}_2 \in \mathbb{R}^{N_2}$, and $\boldsymbol{\theta} \in \mathbb{R}^{N_1+N_2}$ is a concatenated vector.
>
> For example, the functions $g$ and $h$ represent tree ensemble models with a depth of 2 and 3, respectively. Let us assume that both $g$ and $h$ contain the same number of trees and, for simplicity, suppose the case where they exist in a 1:1 ratio. Note that it is easy to generalize it for ensembles containing various tree architectures.
>
> The NTK induced by this model can be decomposed into the sum of the NTKs of each tree architecture as follows:
> $\left\langle \frac{\partial f(\boldsymbol{x}_i, \boldsymbol{\theta})}{\partial \boldsymbol{\theta}},\frac{\partial f(\boldsymbol{x}_j, \boldsymbol{\theta})}{\partial \boldsymbol{\theta}} \right\rangle = \left\langle \frac{\partial g(\boldsymbol{x}_i, \boldsymbol{\theta}_1)}{\partial \boldsymbol{\theta}_1},\frac{\partial g(\boldsymbol{x}_j, \boldsymbol{\theta}_1)}{\partial \boldsymbol{\theta}_1} \right\rangle + \left\langle \frac{\partial h(\boldsymbol{x}_i, \boldsymbol{\theta}_2)}{\partial \boldsymbol{\theta}_2},\frac{\partial h(\boldsymbol{x}_j, \boldsymbol{\theta}_2)}{\partial \boldsymbol{\theta}_2} \right\rangle$
>
> ----
>
> > More interpretation for theorem 3 (e.g., how "theorem 3 tells us that the limiting NTK depends on only the number of leaves at each depth with respect to tree architecture" specifically) can make the importance of theorem 3 clearer as theorem 3 is the first and also the most fundamental contribution of this paper.
>
> Each rule set corresponds to a path to a leaf in a tree, as shown in Figure 2. Therefore, the depth of a rule set corresponds to the depth at which a leaf is present. We have added a description of this interpretation in the revised paper.
>
> ----
>
> > For the interpretation of Figure 3, it's unclear why NTK can be connected with the training behavior and generalization performance of tree ensembles. Any theoretical support for this, like the ones in (Jacot et al., 2018) and [1,2], but with respect to the soft tree ensemble models? I think more explanations need to be provided here since it's an important insight drawn from theorem 3, which also serves as the second contribution shown in the introduction.
>
> (We think you are referring to not Figure 3 but Theorem 3)
>
> Yes. We have theoretical support.
>
> If the limiting NTK does not change during training from its initial values, the training behavior in function space can be understood using kernel regression [1]. For the paper to be self-contained, we have added Theorem 4 in the revised paper, which shows that the limiting NTK induced by an ensemble of arbitrary soft trees does not change during training.
>
> As for the generalization, by using the Rademacher complexity of the kernel regression, its data-dependent generalization bound can be obtained [2, 3], which also applies to our TNTK. This information is also added to Section 2.2 in the revised paper.
>
> ----
>
> > Mentioning "ensemble" in the caption of figure 3 can make this result easier to be understood. There are many lines (w/ different colors) without labels in figures 4 and 6.
>
> We have updated the captions of Figures 3, 4, and 6 in the revised paper. Although the added information was already mentioned in the main text in the initial version of the paper, we have also added it in the captions for better readability as you pointed out.
>
> ----
>
> [1] Jacot et al (2018), Neural Tangent Kernel: Convergence and Generalization in Neural Networks, NeurIPS2018
>
> [2] Bartlett&Mendelson (2002), Rademacher and Gaussian Complexities: Risk Bounds and Structural Results, JMLR
>
> [3] Du et al (2019), Fusing Graph Neural Networks with Graph Kernels, NeurIPS2019

---

> > ### Comment · Reviewer_AJRY · 2022-11-22
> > **Thank you for the response**
> >
> > I thank the authors for the detailed response. Most of my concerns have been addressed especially for the ones that are related to NTK (which I am more familiar with). I would like to keep my score as I am not professional enough in the area of tree ensemble.

---

### Decision · Program_Chairs · 2023-01-20

**Decision:**

Accept: poster

**Justification For Why Not Higher Score:**

The theoretical results in this paper can be of interest to the field of tree ensembles and their applications. All reviewers have positive comments. Not a higher score is because the scale of the audiences in tree ensemble can be relatively smaller than in other machine learning fields.

**Justification For Why Not Lower Score:**

N/A

**Metareview: Summary, Strengths And Weaknesses:**

The paper formulates and analyzes the Neural Tangent Kernel (NTK) induced by soft tree ensembles for arbitrary tree architectures.
This kernel leads to the finding that only the number of leaves at each depth is relevant for the tree architecture in ensemble with infinitely many trees. If the number of leaves at each depth is fixed, the training behavior in function space and the generalization performance is the same across different tree architectures. They also show that the NTK of asymmetric trees like decision lists does not degenerate when they get infinitely deep.

++ The paper is easy to follow. This paper extends the previous study on perfect binary trees to many other soft trees. It can be of wide interest in the field. The theoretical results of this paper are interesting and validated using empirical experiments, which can be helpful in practice.

-- The theoretical analysis of this paper is to an ensemble of the same type of soft trees with the same depth. It's unclear whether the theoretical results in the paper be also applicable to an ensemble of different types of soft trees with various depths, making the paper a bit overclaimed for arbitrary tree architectures, although the authors respond in their rebuttal and the reviewer seems to be satisfied.

The meta-reviewer has carefully read the paper, the reviews, and the authors' responses. It seems that the authors have done a great job in the rebuttal. The meta-reviewer recommends acceptance.


**Note From Pc:**

if the above contains the word "oral" or "spotlight" please see: "oral" presentation means -> notable-top-5% and "spotlight" means -> notable-top-25%. As stated in our emails, we are disassociating presentation type from AC recommendations